# Relationship between El Niño-Southern Oscillation and Atmospheric Aerosols in the Legal Amazon

**Augusto G. C. Pereira** [1,*]**, Rafael Palácios** [1,*]**, Paula C. R. Santos** [2]**, Raimundo Vitor S. Pereira** [1]**, Glauber Cirino** [1] 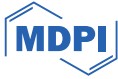 **and Breno Imbiriba** [1]

1    Institute of Geosciences, Federal University of Pará, UFPA, Belém 66075-110, PA, Brazil
2    Institute of Biological Sciences, Federal University of Pará, UFPA, Belém 66075-110, PA, Brazil
*    Correspondence: augusto.pereira@ig.ufpa.br (A.G.C.P.); rpalacios@ufpa.br (R.P.);
     Tel.: +55-(91)-999088462 (A.G.C.P.); +55-(11)-998488650 (R.P.)

**Abstract:** The El Niño-Southern Oscillation (ENSO) stands out as the most significant tropical phenomenon in terms of climatic magnitude resulting from ocean–atmosphere interaction. Due to its atmospheric teleconnection mechanism, ENSO influences various environmental variables across distinct atmospheric scales, potentially impacting the spatiotemporal distribution of atmospheric aerosols. Within this context, this study aims to evaluate the relationship between ENSO and atmospheric aerosols across the entire Legal Amazon during the period from 2006 to 2011. Over this five-year span, four ENSO events were identified. Concurrently, an analysis of the spatiotemporal variability of aerosol optical depth (AOD) and Black Carbon radiation extinction (EAOD-BC) was conducted alongside these ENSO events, utilizing data derived from the Aerosol Robotic Network (AERONET), MERRA-2 model, and ERSSTV5. Employing the Windowed Cross-Correlation (WCC) approach, statistically significant phase lags of up to 4 to 6 months between ENSO indicators and atmospheric aerosols were observed. There was an approximate 100% increase in AOD immediately after El Niño periods, particularly during intervals encompassing the La Niña phase. The analysis of specific humidity anomaly (QA) revealed that, contrary to expectations, positive values were observed throughout most of the El Niño period. This result suggests that while there is a suppression of precipitation events during El Niño due to the subsidence of drier air masses in the Amazon, the region still exhibits positive specific humidity (Q) conditions. The interaction between aerosols and humidity is intricate. However, Q can exert influence over the microphysical and optical properties of aerosols, in addition to affecting their chemical composition and aerosol load. This influence primarily occurs through water absorption, leading to substantial alterations in radiation scattering characteristics, and thus affecting the extinction of solar radiation.

**Keywords:** aerosol; ENSO; black carbon; remote sensing; Amazon

## 1. Introduction

The El Niño-Southern Oscillation (ENSO), in its distinct phases, exerts various modulations on climate due to its atmospheric teleconnection mechanism in the Amazon region [1]. This biome, encompassing 40% of the world's tropical forests [2], is particularly sensitive to the influences of ENSO. Recent studies, such as that by Li et al. [3], demonstrate that climate variations modulated by ENSO can have substantial impacts on atmospheric particles. Additionally, Zhou et al. [4] present two significant perspectives. Firstly, they argue that aerosol emissions resulting from Amazonian fires may interfere with both the intensity and duration of ENSO. Secondly, the study suggests that ENSO might influence the spatial distribution of aerosol emissions originating from Amazonian fires through extreme climatic events, such as severe droughts, potentially leading to outbreaks of fires [5], directly impacting the Amazonian biome [6]. Evaluating aerosol distributions over the Amazon has been increasingly studied, primarily due to their contributions from biomass burning, as in

the works of Alves et al. [7] and Holanda et al. [8,9]. However, investigating the primary hypothesis regarding the relationships with ENSO remains underexplored for the region.

The determination of aerosols is of paramount importance, representing a topic that has gained increasing prominence in research. One of these aerosols of particular significance is Black Carbon (BC), recognized for its unique physical properties. This compound exhibits a strong capacity for solar radiation absorption within the visible and infrared electromagnetic spectrum [10], constituting the most important light-absorbing aerosol in the atmosphere with respect to global warming [11]. BC originates from both incomplete biomass combustion and incomplete fossil fuel burning [12], being emitted both naturally and anthropogenically.

Recent studies, such as those conducted by Holanda et al. [8,9], investigate the impact of long-range transport on aerosol estimates in the Amazon region, with emphasis on BC. The Amazon is subject not only to the influence of fires originating within the biome itself and surrounding areas but is also substantially affected by transatlantic transport from Africa, including smoke, dust, and other particles [9,13]. Lou et al. [14] highlighted the relationship between ENSO and BC emissions in specific regions, resulting in an increase in the frequency of extreme ENSO events, mediated by a series of factors. Among these, the increase in BC emissions from the northern hemisphere stands out, with the potential to attenuate thermal gradients between distinct latitudes and facilitate heat transfer towards the North Pole. Furthermore, such emissions can lead to decreased energy dispersion in the tropics, consequently intensifying the increase in sea surface temperature (SST). The combination of these elements demonstrates the significant contribution of BC to the increase in the incidence of extreme events associated with ENSO. The study by Kim et al. [15] suggests that La Niña shows a direct association with increased aerosol loads in specific regions. This association finds part of its explanation in the moisture transport process, upper tropospheric heating, and increased precipitation during specific periods.

The climatic modulation promoted by ENSO significantly influences the spatiotemporal distribution of atmospheric particles in various distinct regions [16,17]. In the Amazon region, the atmosphere is subject to modulations arising from synoptic systems, exemplified by the Intertropical Convergence Zone (ITCZ), whose seasonal variations manifest from north to south. Additionally, seasonal influences from austral summer systems are also observed, including the reduction in aerosol estimates in the Amazon, attributed to the Bolivian High (BH), Upper Tropospheric Cyclonic Vortex (UTCV), and South Atlantic Convergence Zone (SACZ). The region is also subject to influences from mesoscale convective systems [18] throughout most of the year. The combination of these factors results in pronounced seasonality in aerosol loads [8], as well as in other properties of these particles [19–21].

Within this perspective, a deeper analysis is necessary to comprehend the diverse ways in which climate oscillations, coupled with meteorological elements, may influence estimates of aerosol loads in the Amazon region. Hence, this study aims to assess the spatiotemporal relationship between the ENSO phenomenon and atmospheric aerosols in the Legal Amazon, specifically within the timeframe of 2006 to 2011. Specifically, the study aims to: (1) evaluate the temporal relationship through time series analysis, examining lag times between ENSO indicators and atmospheric aerosol loads; (2) meticulously examine each ENSO period and its potential impacts on atmospheric aerosols loads; and (3) analyze the meteorological conditions associated with these periods. This study resembles a case study, involving the specific assessment of four ENSO events linked to their impacts on aerosol load in the Amazon. Within this brief five-year interval, four ENSO events were identified, revealing a remarkable singularity when compared to prior investigations and the typical average patterns characterizing effects in the Amazon region.

## 2. Materials and Methods

### 2.1. ENSO Indicators Data

In this study, an analysis of sea surface temperature anomalies (SSTA) was carried out in the central tropical Pacific Ocean (CTPO) region from January 2006 to December 2011 (Table 1). Over these four years, two El Niño and La Niña events were identified, alternating between them. The SSTA was evaluated from the western coast of Peru to the eastern region of Melanesia and the eastern coast of Oceania. It is of the utmost importance to meticulously examine ENSO events, as they manifest in two distinct patterns: Modoki/Central Pacific (CP) and Canonical/Eastern Pacific (EP) [22,23].

**Table 1.** Data from the oceanic component, atmospheric component, and atmospheric aerosols used in the study.

| Data | Spatial Resolution | Period | Source | Version | Data Scale |
|---|---|---|---|---|---|
| SSTA in Niño 1+2, Niño 3, Niño 3.4 and Niño 4 regions | $2° \times 2°$ | Availability: 1950–present; Climatology: 1971–2000; Utilized in the Study: 2006–2011. | https://psl.noaa.gov/data/climateindices/list/ (accessed on 9 April 2023) | ERSSTV5 | Monthly |
| SOI | - | Availability: 1951–present; Utilized in the Study: 2006–2011. | https://psl.noaa.gov/data/climateindices/list/ (accessed on 9 April 2023) | - | Monthly |
| $AOD_{500nm}$ (AERONET) | - | Availability: 1993–2021 (AF), 2006–2021 (JP) e 2001–2023 (RB); Utilized in the Study: 2006–2011. | https://aeronet.gsfc.nasa.gov/ (accessed on 12 December 2022) | Nível 2.0 | Monthly |
| $AOD_{550nm}$ extinction for Black Carbon (MERRA2) | $0.5° \times 0.625°$ | Availability: 1980–present; Utilized in the Study: 2006–2011 | https://disc.gsfc.nasa.gov/datasets?project=MERRA-2 (accessed on 27 July 2023) | M2TMNXAER v5.12.4 | Monthly |
| Specific Humidity Anomaly at 2 m (MERRA-2) | $0.5° \times 0.625°$ | Availability: 1980–present; Climatology: 1980–2020; Utilized in the Study: 2006–2011 | https://disc.gsfc.nasa.gov/datasets?project=MERRA-2 (accessed on 20 June 2023) | M2TMNXSLV v5.12.4 | Monthly |
| Vertical Velocity Anomaly (Omega Anomaly, $\omega A$) | $0.5° \times 0.625°$ | Availability: 1980–present; Climatology: 1980–2020; Utilized in the Study: 2006–2011 | https://disc.gsfc.nasa.gov/datasets?project=MERRA-2 (accessed on 18 November 2023) | M2IMNPASM_5.12.4 | Monthly |

Four regions within the CTPO were considered, denoted as Niño 1+2 (80° W–90° W), 3 (90° W–150° W), 3.4 (120° W–170° W), and 4 (150° W–160° E). The Niño 3.4 region is monitored using an index referred to as the Oceanic Niño Index (ONI). Figure 1 illustrates the geographic delineation of the analyzed regions and the SSTA within each Niño region. Oceanic data for monthly anomalies are derived from the Extended Reconstructed Sea Surface Temperature Version 5 (ERSSTV5) dataset, sourced from in situ observations

provided by the Physical Sciences Laboratory (PSL) of NOAA (https://psl.noaa.gov/data/climateindices/list/, accessed on 9 April 2023).

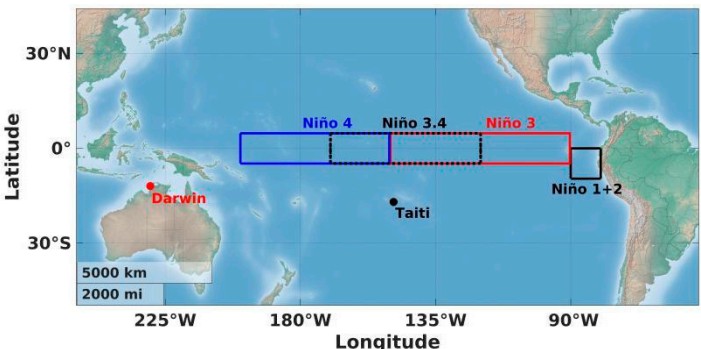

**Figure 1.** Location of the Niño 1+2, 3, 3.4, and 4 regions in the central Tropical Pacific Ocean, where sea surface temperature anomalies were analyzed. The positions of Tahiti and Darwin, used to derive the Southern Oscillation Index (SOI), are also highlighted.

The monthly Southern Oscillation Index (SOI) is derived from data provided by the NOAA Climate Prediction Center (CPC) and involves the analysis of pressure anomalies in Darwin (12.4° S, 130.9° E) and Tahiti (17.5° S, 149° W) in the Southern Tropical Pacific Ocean (Figure 1). Historical data spanning from January 2006 to December 2011 are encompassed in this study (Table 1), and these data are available online at https://psl.noaa.gov/data/climateindices/list/ (accessed on 9 April 2023). During El Niño events (positive ATSM), the SOI exhibits negative values, whereas during La Niña events (negative ATSM), the SOI assumes positive values [24].

*2.2. Atmospheric Aerosol Data*

2.2.1. AERONET

Remote sensing is used to monitor the optical properties of aerosols, primarily through the ground-based monitoring network known as Aerosol Robotic Networks (AERONET). This network continuously provides data on various aerosol parameters and characteristics on a global and regional scale [25]. The parameter utilized in this study is the Aerosol Optical Depth (AOD), which represents the magnitude index of solar radiation extinction caused by aerosols due to scattering and absorption processes [26,27]. AERONET comprises globally distributed spectral radiometers, and AOD data are classified into levels 1.0, 1.5, and 2.0. Level 1 corresponds to raw measurements, level 1.5 includes processed measurements that eliminate cloud and precipitation interferences, while level 2 undergoes final calibration with local corrections and network quality assurance [25].

The sites located within the Legal Amazon, which were used for the assessment of AOD at 500 nm ($AOD_{500nm}$) from level 2.0 data for the period between 2006 and 2011, are derived from Alta Floresta (MT), Ji-Paraná (RO), and Rio Branco (AC) (Table 1). Figure 2 displays the locations of the sites analyzed in this study. These sites are situated in the heart of the Legal Amazon, specifically in the deforestation arc, an area experiencing significant adverse effects from excessive biome exploitation and agricultural expansion.

In the processing of $AOD_{500nm}$ data, we utilized measurements of the Extinction Ångström Exponent (EAE) for the spectral range of 440–870 nm. The EAE provides the spectral dependence of AOD and enables the conversion of $AOD_{500nm}$ to other spectral ranges. Point measurements of $AOD_{500nm}$ and AE from AERONET were employed to calculate daily averages. The AE was used to interpolate AOD from AERONET at 550 nm ($AOD_{550nm}$) in accordance with Equation (1):

$$AOD_{550nm} = AOD_{500nm} \left( \frac{550}{500} \right)^{-EAE} \tag{1}$$

The interpolation of AOD data from 500 nm to 550 nm was conducted with the aim of ensuring coherence and uniformity with the AOD extinction for Black Carbon (EAOD-BC) data, which are within the 550 nm range. This approach enables a more robust analysis of the data and facilitates comparisons between different measurements, thereby enhancing our understanding of the obtained results.

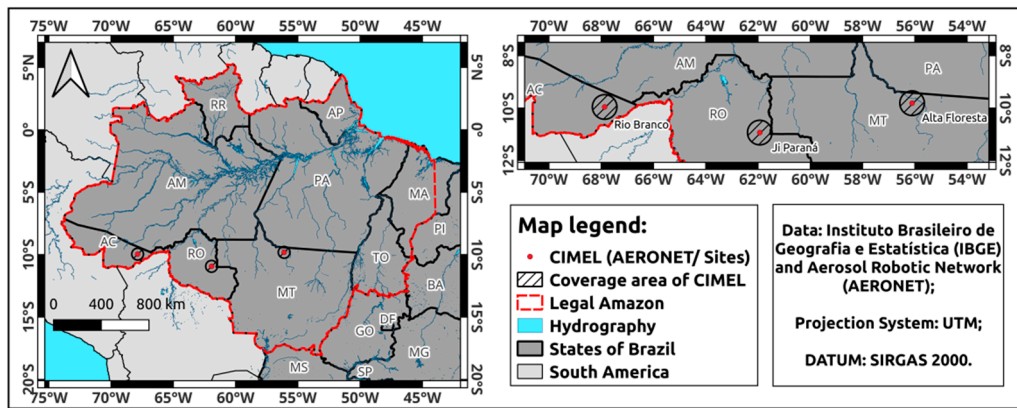

**Figure 2.** Location of the Legal Amazon for spatial evaluation of EAOD-BC and the sites of AERONET in Alta Floresta (state of Mato Grosso, MT), Ji-Paraná (state of Rondônia, RO), and Rio Branco (state of Acre, AC), over which AOD and EAOD-BC data were analyzed, with the location of the CIMEL solar photometer and its coverage area with a radius of 50 km. The Legal Amazon comprises the states of Acre (AC), Amapá (AP), Amazonas (AM), Maranhão (MA), Mato Grosso (MT), Pará (PA), Rondônia (RO), Roraima (RR), and Tocantins (TO).

### 2.2.2. MERRA-2

In pursuit of gaining a deeper understanding of the Black Carbon (BC) load in the Legal Amazon region, we employed the EAOD-BC approach. Unlike AOD, EAOD-BC is a specialized index focused exclusively on BC evaluation. Numerous studies have delved into BC assessment through the absorption mode, as demonstrated in the works of Morais et al. [28] and Dehkhoda et al. [29]. From a different perspective, the study by Rushingabigwi et al. [30] 30 analyzes the extinction mode, resembling the approach utilized in our present study.

To evaluate EAOD-BC over the Legal Amazon, we utilized the Modern-Era Retrospective Analysis for Research and Applications, version 2 (MERRA-2). It is NASA's latest reanalysis system for the satellite era, starting from 1980 [31,32]. This updated system builds upon its predecessor by incorporating more observations and improvements to the Goddard Earth Observing System, version 5 (GEOS-5), produced by the NASA Global Modeling and Assimilation Office (GMAO) [33]. MERRA-2 offers several enhancements over its previous version (MERRA-1), including aerosol assimilation throughout the analysis period [31–33].

The current system refinement takes into account the incorporation and analysis of satellite-derived data, as well as the application of aerosol transport models. Such improvements have a significant impact on the accuracy of BC and other aerosol estimates [32]. Satellite data from the MERRA-2 system include information from the Moderate Resolution Imaging Spectroradiometer (MODIS) sensor to estimate radiation extinction by BC in the atmosphere [31]. For the collection of the time series of EAOD-BC data in Alta Floresta, Ji-Paraná, and Rio Branco, selections were made based on the locations of physical sensors (radiometers) from AERONET between the years 2006 and 2011 (Table 1).

### 2.3. *Specific Humidity Anomaly and Vertical Velocity Anomaly*

In the assessment of Specific Humidity Anomaly (QA) at 2 m above the surface, we computed the climatology using data from the MERRA-2 model (version M2TMNXSLV v5.12.4), spanning the temporal range from 1980 to 2020 (Table 1). Subsequently, monthly

anomaly calculations were performed during the investigation period from 2006 to 2011. To collect the time series of QA data in Alta Floresta, Ji-Paraná, and Rio Branco, selections were made based on the locations of physical sensors (radiometers) from AERONET between the years 2006 and 2011 (Table 1). To examine the vertical velocity anomaly (omega anomaly, $\omega$A), a procedure akin to the one employed for QA was conducted, albeit utilizing version M2IMNPASM_5.12.4 of the MERRA-2 model. A cross-sectional analysis was performed across longitudes and pressure levels in hPa, within the latitude range of $10°$ to $-10°$, encompassing the years from 2006 to 2011 (Table 1).

There is a scarcity of studies conducting a comprehensive evaluation concerning vertical velocity. However, Uma et al. [34] undertook such an assessment by comparing radar-observed data with reanalysis data, including MERRA-2. The study highlights that reducing spatial sampling across all reanalysis data does not yield a significant improvement in the magnitude of vertical velocity. Assessments of directional trends indicate that upward currents are reasonably replicated across all reanalysis datasets, including MERRA-2, yet downward currents are not well captured in the reanalyses [34]. It is important to note that these studies are specific and do not address the particular subject of this article. Nevertheless, they accurately portray uncertainties associated with MERRA-2 concerning vertical velocity.

*2.4. Statistical Methodology*

In the realm of statistical evaluation, we employed the Windowed Cross-Correlation (WCC) technique, which assesses the lagged relationship between time series, taking into account the periodic characteristics of these time series. As described by Boker et al. [35], the WCC method is capable of quantifying associative variability. Within this context, WCC should track changes in the temporal lag (also known as delay) and the strength of association between both time series throughout the experiment (sampling).

With some limitations, the method aims to estimate the time interval between measurements where the maximum association occurs between the two time series and the strength of this association. The importance of finding the optimal association delay has been emphasized in previous studies, such as Zhang et al. [36]. The method should provide an estimate of the temporal delay between the event in $X$ $\{X_1, X_2, \ldots, X_N\}$ and the event in $Y$ $\{Y_1, Y_2, \ldots, Y_N\}$, as well as the strength of the association between the two events. In the statistical analysis, we considered the time series from the Niño 3.4 region, which effectively represents the distinct phases of ENSO. Additionally, we used the EAOD-BC and $AOD_{550nm}$ series for all three sites. To address gaps in missing $AOD_{550nm}$ data, we opted to apply linear interpolation to these series. The WCC between two time series $X$ = Niño 3.4 and $Y$ = Atmospheric Aerosols (AOD and EAOD-BC) with a delay $\beta$ is given by Equation (2):

$$WCC(X, Y, \beta) = \frac{1}{N - \beta} \sum_{i=1}^{N-\beta} \frac{(x_i - \overline{X})(y_{i+\beta} - \overline{Y})}{std(X)\, std(Y)} \tag{2}$$

where $\overline{X}$ and $\overline{Y}$ represent the overall means, and $std(X)$ and $std(Y)$ represent the standard deviations of $X$ and $Y$, respectively, across all measurement instances. This method is essentially a standard Pearson correlation between the two time series lagged by observations.

## 3. Results

*3.1. Time Series Analysis*

Figure 3 illustrates the temporal variability of the ENSO indicators during the period from 2006 to 2011 in the Niño regions (Figure 3a–d) and the SOI (Figure 3e), situated in the Pacific Ocean. Within this temporal span, four ENSO events were identified, comprising two El Niños and two La Niñas. The first El Niño, depicted in red, occurred in the years 2006 and 2007, while the second one took place between 2009 and 2010. In contrast, the La Niña periods, highlighted in blue, spanned from 2007 to 2008 and from 2010 to 2011. Segments

where no prominent red or blue highlights are observed represent periods of neutrality. The identification of ENSO events was based on the five time series in Figure 3, in conjunction with the widely adopted classifications found in the studies of Trenberth [24,37] and Cai et al. [22], along with the ONI index available at https://psl.noaa.gov/data/climateindices/ list/ (accessed on 9 April 2023).

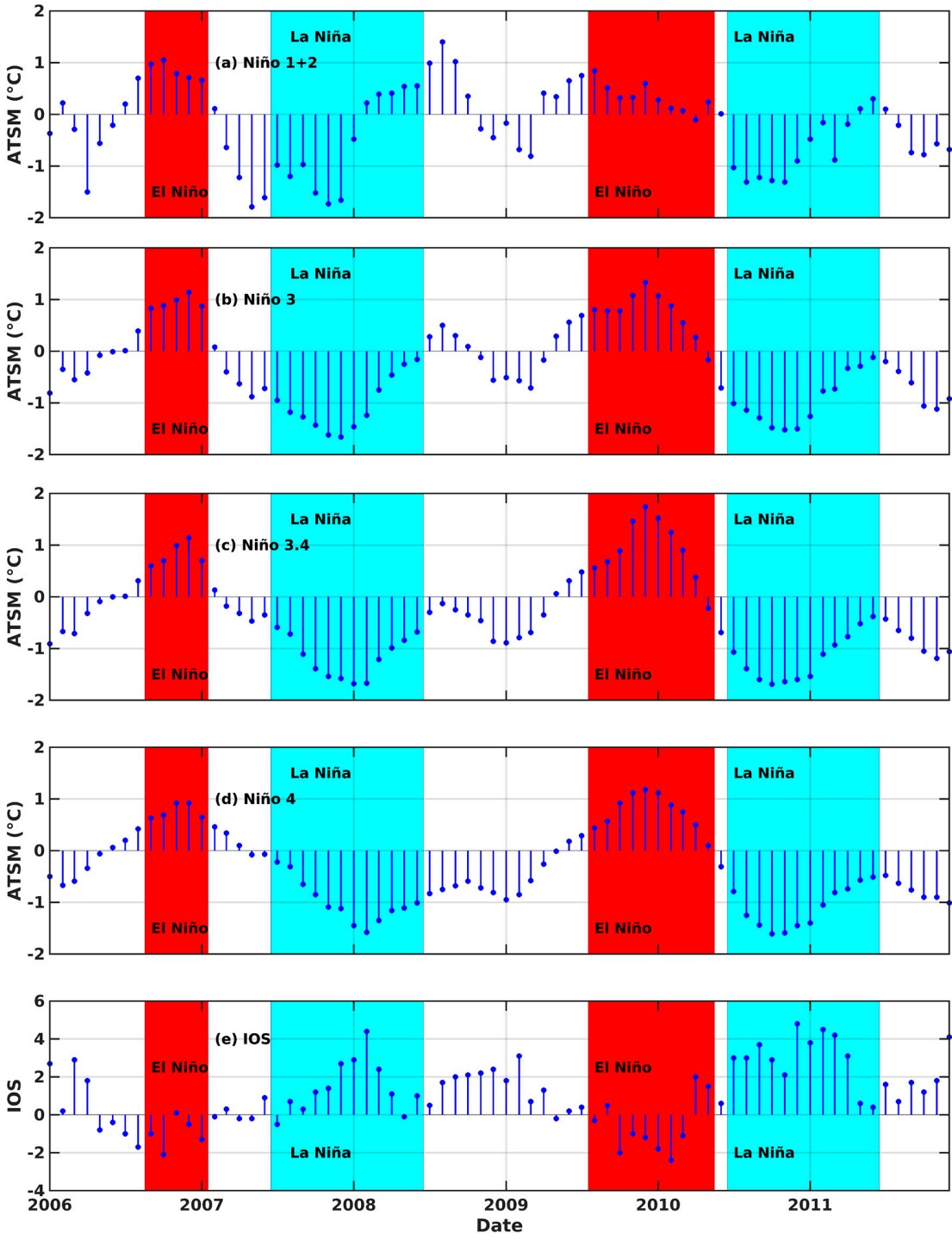

**Figure 3.** Time series corresponding to ENSO indicators, sea surface temperature anomalies (SSTA) over Niño regions (**a**) 1+2, (**b**) 3, (**c**) 3.4, and (**d**) 4, and (**e**) Southern Oscillation Index (SOI). The highlight in red at the background of the chart represents El Niño periods, while blue indicates La Niña periods.

The time series of the $AOD_{550nm}$ from the AERONET is depicted in Figure 4 for the sites in Alta Floresta, Ji Paraná, and Rio Branco. All sites exhibit a noticeable seasonality during the period spanning from 2006 to 2011, as expected, due to significant emissions resulting from fires that occur during the dry season [28,38]. Sites located in the Legal Amazon region manifest distinct variability characteristics throughout the year, highlighting both a wet season and a dry season. The first half of the year is classified as the wet season, whereas the second half is characterized as the dry season, a categorization previously established by [28,39]. As depicted in the supplemental material, Figure S1 presents the year-to-year variability of AOD on a monthly scale, alongside the respective average and standard deviations. Additionally, Figure S2 illustrates the occurrence of fire outbreaks within the states encompassing the specific locations addressed in this study. These visual representations facilitate a more comprehensive understanding of the relationship between fire outbreaks and the variability in atmospheric aerosol loads over the Amazon region. It is evident that the AOD closely tracks the temporal trajectory of fire outbreaks, signifying their significant influence on the fluctuation of these aerosols.

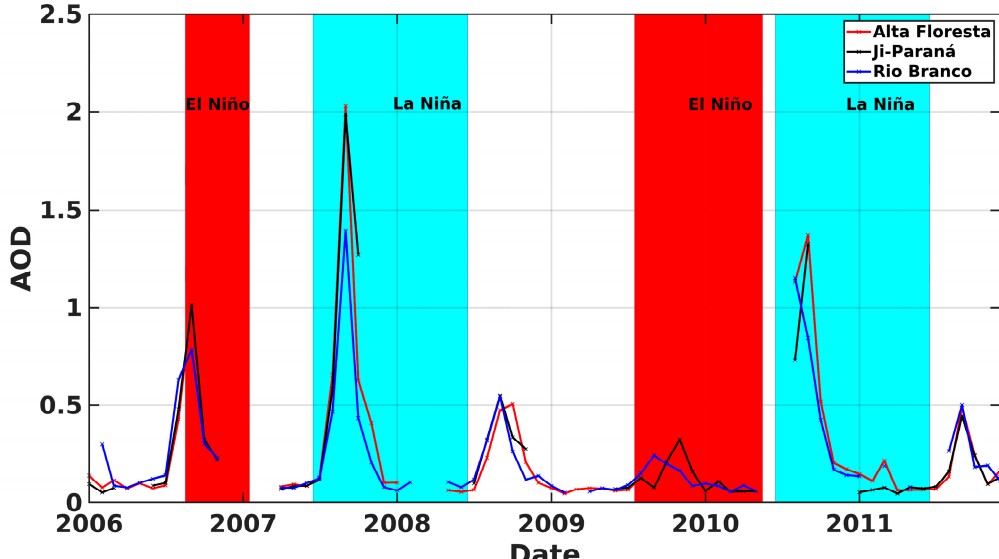

**Figure 4.** Time series of aerosol optical depth (AOD) at 550 nm from AERONET with monthly averages for the sites of Alta Floresta, Ji-Paraná, and Rio Branco. The highlight in red at the background of the chart represents El Niño periods, while blue indicates La Niña periods.

During the wet season, natural and anthropogenic emissions prevail, whereas during the dry season, due to emissions from both natural and anthropogenic fires, atmospheric aerosol loads increase significantly, resulting in elevated AOD estimates [40]. The time series exhibit substantial data gaps during the wet season, attributed to the high cloud cover and precipitation in the Amazon region.

Similar to AOD estimates, which provide a broader view of aerosol loads, Figure 5 illustrates the exclusive temporal variability of EAOD-BC, derived from the MERRA-2 model, for the locations of Alta Floresta, Ji Paraná, and Rio Branco.

The EAOD-BC exhibits an annual variability similar to that of AOD, displaying seasonality in both the dry and wet seasons. It is worth emphasizing that EAOD-BC estimates bear a remarkable resemblance to AOD. During the wet season, EAOD-BC values hover around 0.01, while in the dry season, higher values are observed, reaching approximately 0.15 to 0.2 for Alta Floresta. Holanda et al. [8] underscore that the Amazon biome harbors a significant load of BC derived from emissions resulting from biomass combustion, encompassing both forest fires and burnings employed for deforestation and agricultural activities. In this context, these anthropogenic actions release substantial quantities of particulate matter into the atmosphere, with BC being one of the predominant elements among atmospheric aerosols in the region under study.

According to Morais et al. [28], in the deforestation arc region BC contribution is dominant during the dry season. To some extent, all locations in the deforestation arc exhibit aerosol loads with a mixture of dust, BC, and Brown Carbon (BrC). These values possibly result from the long-distance (intercontinental) transport of Saharan and regional aerosol plumes, which initially arrive in the Alta Floresta region and subsequently move towards the Ji-Paraná and Rio Branco regions [8]. Regarding transportation, particularly from August to October, long-distance transport plays a significant role in influencing the load of BC in the Amazon [21].

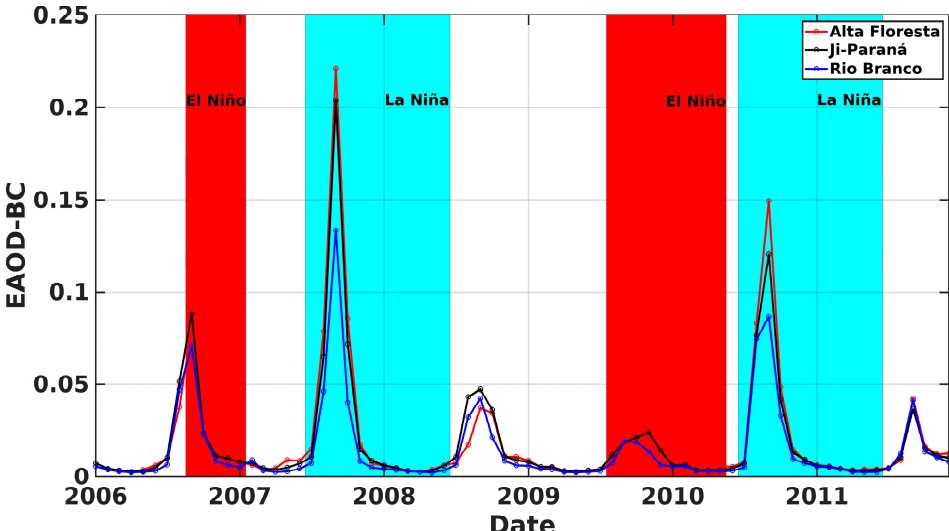

**Figure 5.** Time series of aerosol optical depth extinction for Black Carbon (EAOD-BC) at 550 nm with monthly averages for Alta Floresta, Ji-Paraná, and Rio Branco. The highlight in red at the background of the chart represents El Niño periods, while blue indicates La Niña periods.

### 3.2. Windowed Cross-Correlation

Utilizing the WCC (Figures 6 and 7) allows for a more nuanced understanding of the dynamic interactions between the time series, enabling the identification of lags and contributing to a precise data analysis and interpretation. Figures 6 and 7 illustrate the temporal relationship between the ENSO indicator, the SSTA in the Niño 3.4 region, and the AOD estimates (Figure 6) and EAOD-BC (Figure 7) during the period from 2006 to 2011. We will examine negative lags, which represent the delay of AOD and EAOD-BC estimates concerning the ENSO indicator time series.

In general, all the conducted analyses have revealed a consistent relationship between the ENSO indicator and atmospheric aerosols, displaying a consistent pattern of lag and cross-correlation coefficients (ρCC) throughout the entire analysis period at the three sites. Observing Figures 6 and 7 during the analysis period, ρCC oscillated between positive and negative values. Between 2007 and 2008, negative ρCC values (approximately −0.7) were identified with lags of up to 6 months, indicating that the time series were out of phase during this period. Increases (or decreases) in the ENSO indicator led to decreases (or increases) in AOD and EAOD-BC estimates.

Conversely, between 2007 and 2009, positive ρCC values were observed, also with a lag of up to 6 months, suggesting that the time series were in phase during this period. Increases (or decreases) in the ENSO indicator resulted in increases (or decreases) in AOD and EAOD-BC estimates. It is noteworthy that during this period, for AOD in Ji-Paraná (Figure 6b) and Rio Branco (Figure 6c), some fluctuations in ρCC were observed, along with lower ρCC values, around 0 to 0.5, indicating a lack of statistical significance.

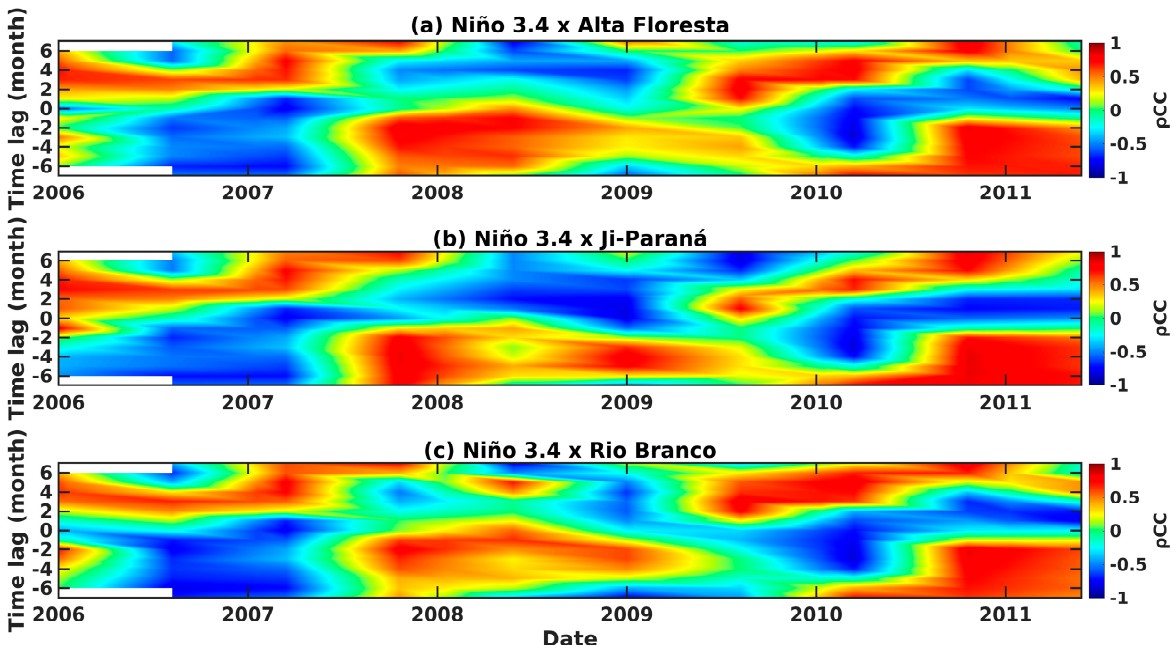

**Figure 6.** Windowed cross-correlation (WCC) analysis between the time series of SST anomalies (SSTA) in the Niño 3.4 region and the time series of aerosol optical depth (AOD) at 550 nm for (**a**) Alta Floresta, (**b**) Ji-Paraná, and (**c**) Rio Branco. In this analysis, a positive ρCC indicates that the Niño 3.4 ENSO SSTA series are in phase with the AOD, while a negative ρCC indicates that they are out of phase.

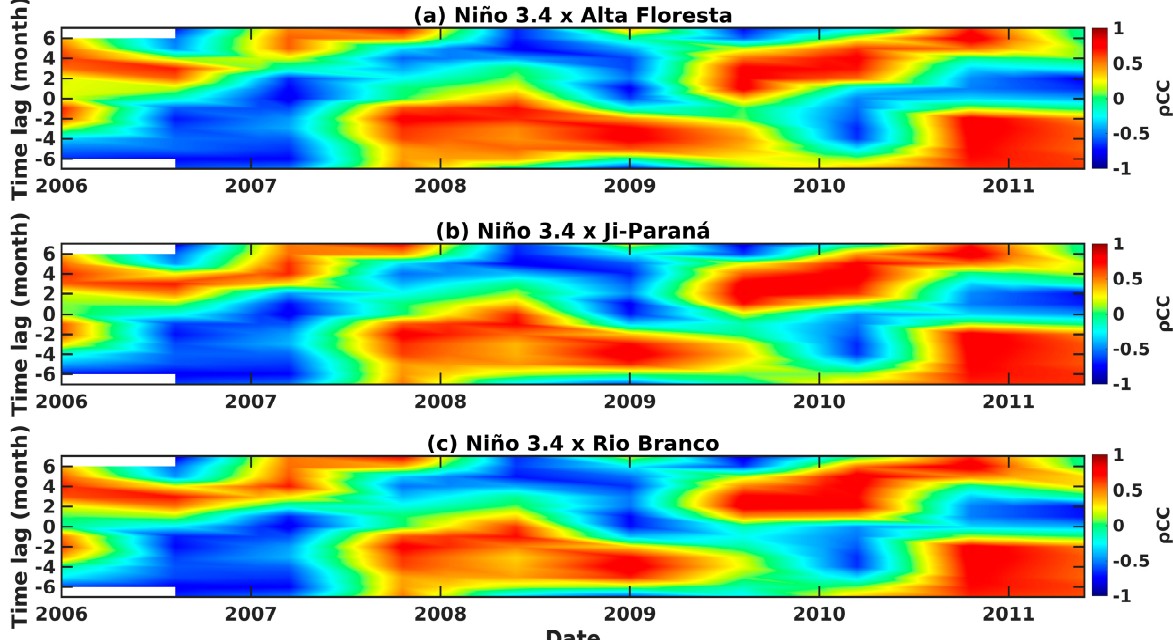

**Figure 7.** Windowed cross-correlation (WCC) analysis between the time series of SSTA (SSTA) in the Niño 3.4 region and the time series of aerosol optical depth extinction for Black Carbon (EAOD-BC) at 550 nm for the sites of (**a**) Alta Floresta, (**b**) Ji-Paraná, and (**c**) Rio Branco. In this analysis, a positive ρCC signifies that the Niño 3.4 ENSO SSTA series are in phase with the EAOD-BC, whereas a negative ρCC suggests they are out of phase.

Towards the end of 2009 until 2011, both negative and positive ρCC were found for all analyses, with positive ρCC predominantly during this period. Between 2009 and 2010, a lag of up to 4 months was observed between the ENSO indicator and atmospheric aerosols,

with ρCC approximately −0.8 for AOD and approximately −0.6 for EAOD-BC (Figure 7), both statistically significant. During this period, increases (or decreases) in the ENSO indicator resulted in decreases (or increases) in AOD and EAOD-BC. Hence, atmospheric aerosols were out of phase with the ENSO indicator.

In the period from 2010 to 2011, ρCC predominantly displayed positive values, exceeding 0.65. These values were observed for the locations of Alta Floresta (Figure 6a) and Ji-Paraná (Figure 6b), reaching approximately 0.75, with higher statistical significance. Until the middle of 2010, regarding EAOD-BC (Figure 7), there was not as much statistical significance in the ρCC value compared to the AOD analysis until the middle of 2010. However, concerning AOD, a lag ranging from 5 to over 6 months in relation to the ENSO indicator was observed until the middle of 2010. From the middle of 2010 until the end of the series, higher ρCC values were observed, which stood out as the maximum values throughout the entire analysis, with lags of up to 6 months. Consequently, it was found that increases (or decreases) in the ENSO indicator resulted in increases (or decreases) in atmospheric aerosols, indicating that the time series were in phase with a lag of up to 6 months.

### 3.3. Spatiotemporal Assessment

Through a meticulous examination of the time series of ENSO indicators (Figure 3), along with AOD estimates (Figure 4) and EAOD-BC (Figure 5), and further delving into this assessment using WCC (Figures 6 and 7), it is possible to identify two similar scenarios during the El Niño and La Niña periods. In both cases, El Niño periods were evident: the first occurred between September 2006 and January 2007 (EN-0607), lasting for five months (Figure 8a); the second spanned from August 2009 to May 2010 (EN-0910), encompassing a duration of ten months (Figure 9a).

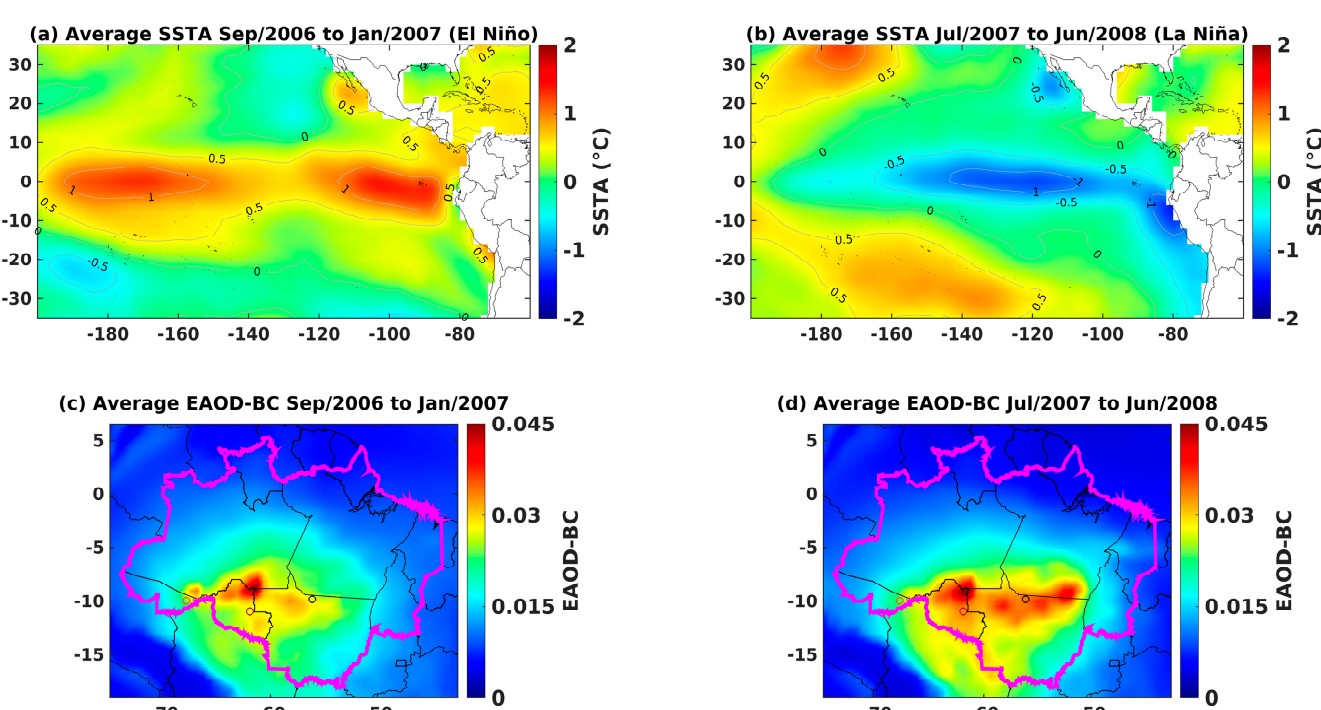

**Figure 8.** Evaluation over the Pacific Ocean showing (**a**) average SSTA during the period September/2006 to January/2007, covering the El Niño period, (**b**) average SSTA during the period July/2007 to June/2008, covering the La Niña period, and average spatial distribution of extinction aerosol optical depth extinction for Black Carbon (EAOD-BC) at 550 nm during the periods (**c**) El Niño and (**d**) La Niña.

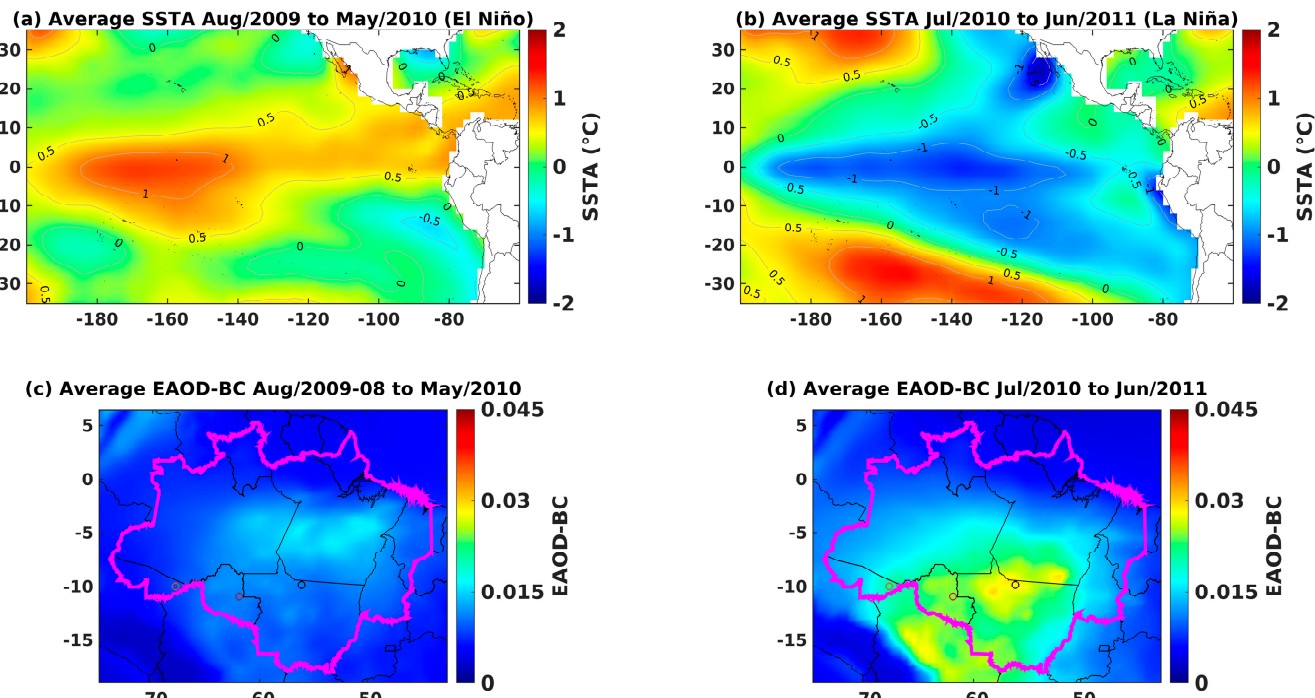

**Figure 9.** Evaluation over the Pacific Ocean showing (**a**) average SSTA during the period August/2009 to May/2010, covering the El Niño period, (**b**) average SSTA during the period July/2010 to June/2011, covering the La Niña period, and average spatial distribution of aerosol optical depth extinction for Black Carbon (EAOAD-BC) at 550 nm during the periods (**c**) El Niño and (**d**) La Niña.

Following these two El Niño periods, La Niña years were identified, each lasting for 12 months: one between July 2007 and June 2008 (LN-0708), and another between July 2010 and June 2011 (LN-1011). During these La Niña periods, there was a significant increase in AOD, as observed in the time series (Figure 4), as well as an increase in EAOD-BC, as evidenced in the time series (Figure 5), with detailed spatial analyses of the Legal Amazon region illustrated in Figures 8c,d and 9c,d.

In the analysis conducted through WCC, statistical significance was found between the ENSO indicator and atmospheric aerosols. The ρCC reached its minimum values between the years 2006 and 2007, with a delay of approximately 6 months. After the increase in AOD and EAOD-BC estimates throughout the year 2006, reductions in SST were observed in all Niño regions, with a lag of approximately six months. The ENSO phase exhibited an opposite phase to that of atmospheric aerosols during the six-month period between 2006 and 2007, supporting the perspective of an increase in atmospheric aerosols in LN-0708 as a result of the EN-0607 event. This relationship becomes evident when analyzing EAOD-BC estimates in the Legal Amazon region, where significant increases in EAOD-BC were observed, especially at the locations of the three AERONET network sites, as demonstrated in Figure 8c,d.

Examining oceanic characteristics during the EN-0607 (Figure 8a) and LN-0708 (Figure 8b) periods, in conjunction with variations in EAOD-BC estimates over the Legal Amazon region between these two events, a considerable increase in EAOD-BC estimates over the central-southern region of the Legal Amazon was observed. The border region comprising the states of Rondônia, Amazonas, and Mato Grosso remained stable in terms of estimates. However, on a broader scale, in the central-southern portion of the biome, a significant increase was noted after the EN-0607 period, during the LN-0708 coverage. The time series (Figures 4 and 5) revealed increases of approximately 100% in the peaks of AOD and EAOD-BC between the intervals of these events. In the northwest, north, and northeast regions of the biome, conditions remained consistent, with low EAOD-BC estimates and no significant changes.

As highlighted by Cai et al. [22], the year 2007 was categorized as a CP/Modoki La Niña event. The authors suggest that the onset of this specific La Niña phenomenon (during the months of June, July, and August) implies positive anomalies in air temperatures in the area covered by the deforestation arc, precisely where significant increases in EAOD-BC were observed. This atmospheric condition can be identified as a determining factor for the increase in atmospheric aerosol estimates, particularly EAOD-BC, as forest fires become more recurrent and susceptible during this period due to abnormal air temperature variations. Consequently, there is an intensification in aerosol and pollutant emissions into the atmosphere.

Between the years 2009 and 2011, a similar situation to the previous one was observed, as illustrated in Figure 8. This scenario was characterized by an increase in EAOD-BC estimates after an El Niño period, followed by a La Niña event. It is crucial to highlight the disparities in the prevailing oceanic conditions between these periods. Comparing Figures 8 and 9, both the EN-0607 event (Figure 8a) and the LN-1011 event (Figure 9b) manifested more strongly, as evidenced by SSTA in the Pacific Ocean, unlike the EN-0910 (Figure 8a) and LN-0708 (Figure 7b) events, which occurred more weakly in terms of SSTA. For EAOD-BC estimates, lower values were obtained in these latter two ENSO events (Figure 9c,d), as well as in the time series of AOD and EAOD-BC for the three sites.

It is worth noting that during the ENOS episodes that occurred between 2006 and 2008, SST anomalies exhibited different distributions between the central and coastal regions of the Pacific Ocean. In contrast to these ENOS events, which occurred between 2009 and 2011, a concentration of higher SSTA was observed for EN-0910 and lower SSTA for LN-1011 in the central portion of the Pacific Ocean. This analysis assumes primary relevance due to the ocean atmosphere interaction, which presumably exerted influence on atmospheric aerosol estimates over the Legal Amazon region.

In the period from 2009 to 2011, $\rho CC$ values were found to oscillate between positive and negative, with a maximum lag of approximately 4 to 6 months between the ENSO indicator and atmospheric aerosols. Similar to the analysis represented in Figure 8, there was an increase of over 100% in AOD and EAOD-BC peaks during the interval corresponding to the occurrence of El Niño and La Niña phenomena (ENSO).

Regarding EAOD-BC in the Legal Amazon region, during the EN-0910 event (Figure 8a), the most significant estimates were observed in the areas east of the Amazon and in the central part of Pará. Subsequently to this event, during the LN-1011 period (Figure 9b), changes in the distribution of EAOD-BC (Figure 9c,d) occurred across virtually the entire extent of the Legal Amazon. It is worth noting in this context that the central-southern portion of the biome presented the most significant estimates, with Alta Floresta and Ji-Paraná particularly more affected compared to Rio Branco. Similar to Figure 8, in Figure 9, in the deforestation arc area where the three sites are located, they are evidently the most impacted by EAOD-BC.

In all four periods analyzed, the state of Rondônia consistently exhibits EAOD-BC values above 0.025 across virtually its entire area, reaching approximately 0.045 in some periods. These values are considerably higher compared to other states. The state of Mato Grosso presents a situation similar to that of Roraima, with the northern part of the state being significantly affected. In time series analyses, this trend becomes more evident, especially through AOD, which provides a more precise representation of atmospheric aerosol loads. The southern regions of the Amazonas and Pará states were the most impacted due to their location, as they are part of the deforestation arc. On the other hand, the states of Roraima, Tocantins, and Maranhão showed less pronounced influence of EAOD-BC in the four periods analyzed.

It is essential to conduct a detailed analysis of the two scenarios presented, in which the decrease in SST resulted in La Niña events. As a result, there was an increase in AOD and EAOD-BC estimates, which is an unusual situation, since in La Niña years, meteorological and environmental conditions typically favor a decrease in atmospheric aerosol estimates in the Legal Amazon region, where the study is situated.

## 4. Discussion

According to Nascimento and Senna [41], the scenario illustrated in Figure 9 can be attributed to the La Niña event being classified as CP/Modoki. This classification entails an atmospheric teleconnection mechanism that influences meteorological variables, resulting in an increased risk of wildfires. As for the case depicted in Figure 8, the assessment conducted by Palácios et al. [38] revealed that the year 2007 exhibited the highest number of wildfire incidents between 2002 and 2017. During this period, variations in atmospheric aerosols directly correlate with the surge in wildfire occurrences, with wildfires serving as a major source of aerosol emissions within the study area. In accordance with the findings of Barbosa et al. [42], there was a notable uptick in the incidence of wildfires and affected areas during La Niña events. The authors indicate that meteorological factors, in conjunction with social and environmental aspects, play a pivotal role in influencing these variables.

In this context, comprehending these two scenarios can be grounded in the study by Silva Junior et al. [43], which emphasizes that stronger trade winds have the potential to propagate wildfires, a phenomenon exacerbated during La Niña periods. The circulation of trade winds originating from the northeast exhibits greater velocity, thereby accelerating the dynamics of wildfires in the region. It is crucial to highlight that trade winds exert a significant impact on the occurrence and spread of wildfires, provided favorable conditions of temperature and relative humidity exist. These winds, renowned for their regularity and constancy, can expedite the combustion of biomass and facilitate the rapid spread of fires across extensive areas. In summary, trade winds play a fundamental role in transporting particles and act as facilitators for the expansion of fires into adjacent regions.

### 4.1. The Walker Circulation

Through the analysis of the vertical velocity anomaly (omega anomaly, $\omega A$), as shown in Figure 10, it becomes possible to comprehend the underlying physics behind the differences identified in this study. These anomalies elucidate the variations in the Walker circulation during periods associated with ENSO events, particularly concerning differences from the average climatological patterns. The average vertical velocity ($\omega$ omega) for each period is presented in the supplementary materials (Figures S3–S6). In the cross-section representation displayed in Figure 10, the Pacific Ocean is depicted between $-180°$ and $-80°$, the Legal Amazon region spans approximately $-75°$ to $-45°$, and the sites range between $-70°$ and $-55°$. The differences influenced by the ENSO phases are clearly observable.

During the EN-0607 (Figure 10a) and EN-0910 (Figure 10c) periods, negative $\omega A$ anomalies prevail over the Pacific Ocean, intensifying convective capacity (ascending branches of the Walker circulation). However, the opposite is observed over the Legal Amazon, with positive $\omega A$, resulting in decreased convective capacity. In certain parts of the atmospheric column, negative $\omega A$ values are noted, such as at approximately $-80°$ and $-60°$ longitudes (Figure 10a) during EN-0607. For EN-0910, closer to the lower atmospheric levels shown in Figure 10c, prevalent positive values are also observed, favoring cloud formation at lower levels.

Observation sites in Alta Floresta, Ji Paraná, and Rio Branco are more affected by aerosol loads during La Niña periods, LN-0708 (Figure 10b), and LN-1011 (Figure 10d). Consequently, a predominance of negative $\omega A$ is noted in the eastern sector of the Amazon, while in the central and western regions, positive $\omega A$ is observed, mainly at lower levels where humidity is more prevalent. These conditions result in a reduced propensity for convective activities, significantly affecting the formation of precipitating clouds and favoring the emission of atmospheric aerosols due to atmospheric conditions.

Between 2006 and 2008, the $\omega A$ conditions were highly favorable for aerosol emissions in the Amazon, reflected in the high values of AOD and EAOD-BC observed in these time series compared to other years. Specifically, during the LN-0708 event, the highest values of AOD and EAOD-BC were recorded. The Walker circulation can partly explain this phenomenon, as its subsiding cell was more prominent between $-180°$ and $-60°$,

encompassing a large part of the Amazon and creating atmospheric conditions conducive to dryness and drought.

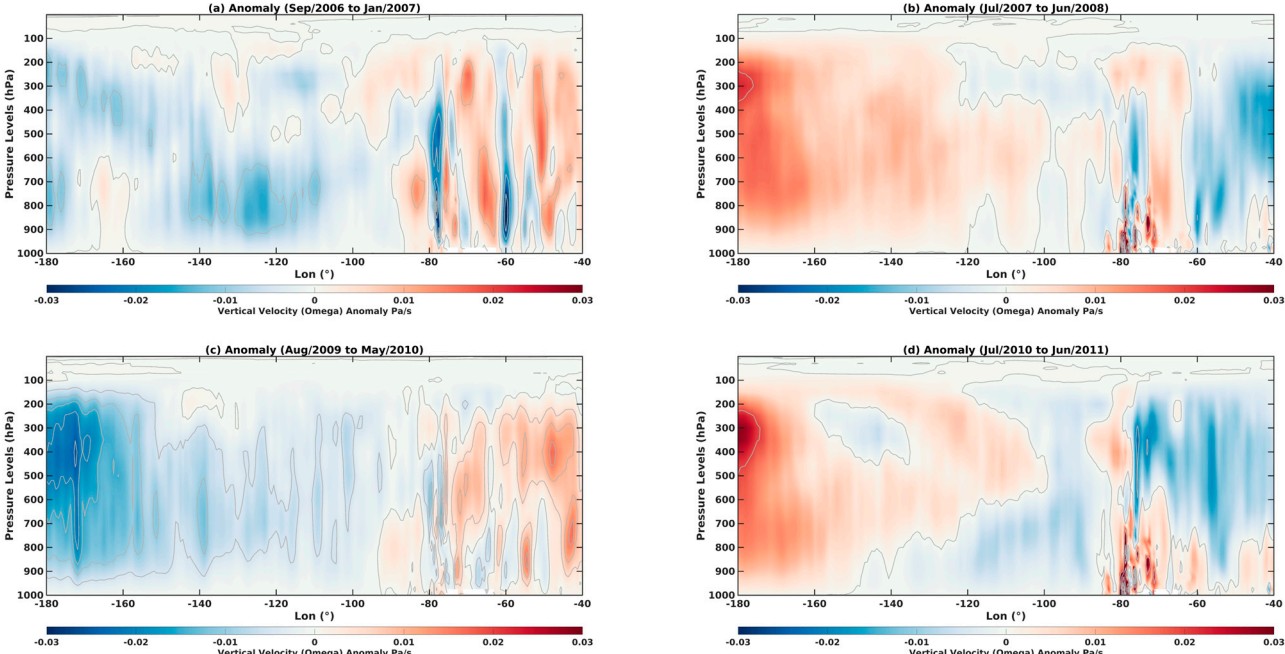

**Figure 10.** The cross-section analysis between longitude and pressure levels (hPa) within the latitudinal range of 10° and −10° showcases anomalies in vertical velocity (ω anomaly, ωA) measured in Pa/s for the following periods: (**a**) Sepember/2006 to January/2007 encompassing the EN-0607 event, (**b**) July/2007 to June/2008 encompassing the LN-0708 event, (**c**) August/2009 to May/2010 encompassing the EN-0910 event, and (**d**) July/2010 to June/2011 encompassing the LN-1011 event. Negative values of ωA indicate intensified upward activities, whereas positive values of ωA signify intensified subsiding activities.

The lag relationships of 4 to 6 months, identified by the WCC, can be elucidated through the study by Kumar and Hoerling [44]. This study addresses the lagged relationship between the Walker circulation and tropical atmospheric responses. The observed lag in the tropical atmospheric response is associated with the persistence or extension of this response after the peak of the austral summer (boreal winter), specifically in relation to SSTA in the Niño 3.4 region. This indicates that the influence of the Walker circulation continues to affect the tropical atmospheric response for an extended period after the initial peak, evidenced by the persistence of this response for up to 9 months after the winter peak.

Additionally, there is an interval of 1 to 3 months between the peak in tropospheric height response at 200 hPa and the peak in SSTAs in the Niño 3.4 region, located in the eastern tropical Pacific [44]. Furthermore, these lags are possibly associated with the concept proposed by Alexander [45] known as "the atmospheric bridge", where ENSO presents an influencing mechanism on the Atlantic Ocean temperatures and, consequently, on meteorological elements through this remote interaction, the teleconnections. In this context, there are delays associated with the Walker circulation. Thus, in the North Atlantic, the reduction in trade winds, between approximately 5° and 20° latitude, decreases the latent heat flux to the atmosphere during the austral summer and boreal autumn. The latent heat flux represents the amount of heat transferred from the sea surface to the atmosphere through water evaporation. With the decrease in trade winds, evaporation decreases, resulting in less heat transfer from the sea surface to the atmosphere. This scenario can cause warming of the subtropical North Atlantic surface in spring since less heat is dissipated to the atmosphere [45]. This sector of the Atlantic Ocean is highly

important for the Amazon atmosphere, as the northeast trade winds originating from the North Atlantic enter the Amazon.

### 4.2. Specific Humidity Assessment

In order to delve deeper into the preceding analyses, aiming to comprehend the periods of ENSO influence and its potential concurrent effects with meteorological characteristics in the Legal Amazon region, Figure 11 depicts the variability of specific humidity (Q) and the QA over the temporal span from 2006 to 2011 for three distinct locations: Alta Floresta (Figure 11a), Ji Paraná (Figure 11b), and Rio Branco (Figure 11c).

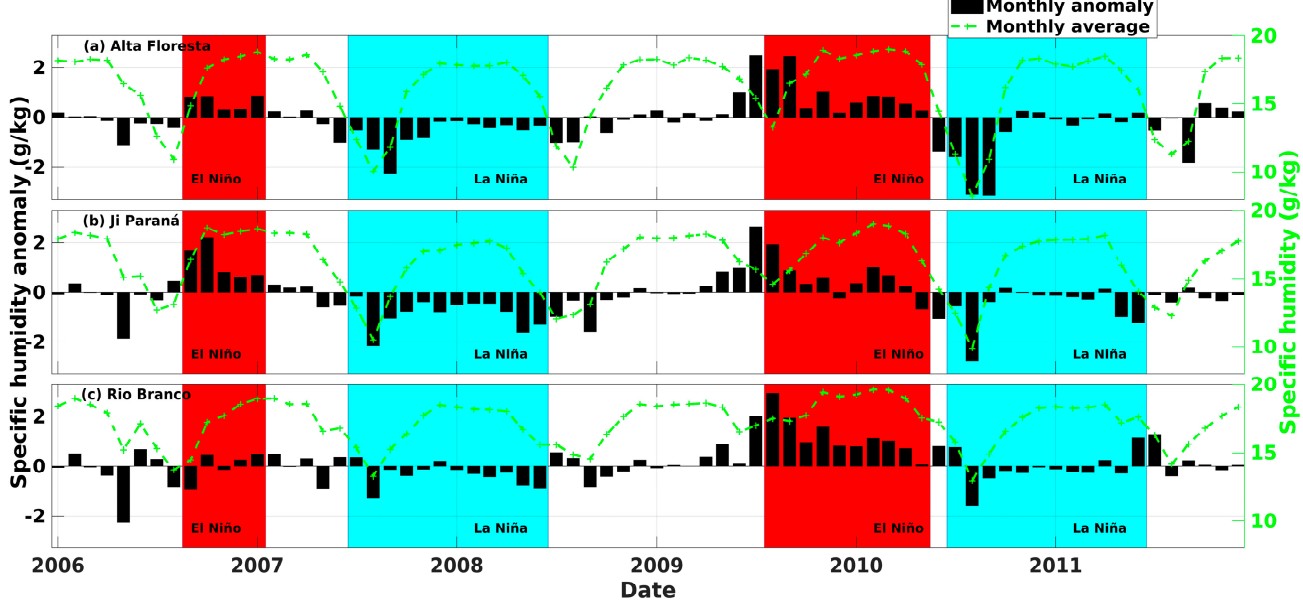

**Figure 11.** Time series of MERRA-2 model data between 2006 and 2011 for specific humidity (g/kg) on the dashed green lines and specific humidity anomaly (g/kg) on the black bars, based on the climatology between 1980 and 2020, both series based on the locations of AERONET sites. The red and blue highlights indicate the periods of El Niño and La Niña, respectively.

Considering the strong relationship between humidity, ENSO, and aerosols, as indicated by Zeng et al. [46], it is observed that humidity significantly influences the composition, load, concentration, and hygroscopic properties of aerosols. This factor plays a crucial role in both the formation and exacerbation of aerosol pollution. Additionally, as highlighted by Gushchina et al. [47], a study conducted in the tropical region of Earth concluded that ENSO impacts humidity in various tropical areas. During El Niño events, regions such as Indonesia, Australia, and parts of Africa and South America tend to become drier, while areas like western Australia, eastern Africa, and the southern part of South America become wetter. In La Niña periods, these patterns typically oppose those observed during El Niño. The indices of drought vary depending on the type of ENSO (Modoki/Central Pacific and Canonical/Eastern Pacific) and exhibit complex relationships with atmospheric anomalies.

During the EN-0607 and EN-0910 episodes, positive values of QA prevail across all sites, with Q levels exceeding 14 g/kg. It is worth noting that during these El Niño periods, higher SH rates are observed over the biome. Importantly, these El Niño events coincide with the phase categorized as the dry season of atmospheric aerosols. In these two events, the lowest estimates of AOD and EAOD-BC were observed compared to subsequent La Niña events. Conversely, during the LN-0708 and LN-1011 episodes, QA predominantly exhibited negative values, especially at the onset of each event. Remarkably, for Rio Branco (Figure 11c), elevated Q and QA were observed, conditions conducive to obtaining the lowest estimates of AOD and EAOD-BC.

The El Niño and La Niña scenarios discussed here stand out for their notable uniqueness when contrasted with widely discussed studies and average patterns characterizing events affecting the Amazon region. According to Foley et al. [48], in the Amazon region, El Niño episodes are associated with warmer and drier meteorological conditions than the usual pattern. Conversely, La Niña events bring cooler and moister environmental conditions. Focusing on the hydrological aspect, variations resulting from ENSO influence both soil and air humidity in the Amazon, mediated by alterations in rainfall rates. However, Withey et al. [49] emphasize the need to incorporate forest fire occurrences associated with El Niño when estimating carbon emissions from the Amazon region. In this perspective, substantial differences were identified in this study, resulting in increases in atmospheric aerosol estimates during La Niña episodes and the opposite trend for El Niño events. Thus, by examining changes over time in Q and QA, in conjunction with ENSO climate patterns, it becomes possible to recognize complex and significant connections between climatic events and atmospheric conditions in the Legal Amazon.

According to Sousa et al. [50], it is plausible to infer that El Niño exerts a substantial influence on subsequent precipitation rates in specific areas of the Amazon region. This observation aligns with the analysis by Souza et al. [51], who, through statistical analysis, demonstrate a statistically significant relationship between atmospheric aerosols and precipitation.

In this context, the temporal discrepancies detected, as evidenced through the WCC, between ENSO events and the distribution of AOD and EAOD-BC in the Legal Amazon, could possibly be attributed to changes in rainfall patterns in the studied region. However, such temporal gaps may also be associated with the QA identified within the scope of the region under examination.

*4.3. Aerosol Interaction*

The conditions influenced by ENSO oscillation, assessed through meteorological parameters such as Q, QA, and atmospheric circulation using ωA to investigate the Walker circulation, offer substantial support to the prospect that both La Niña events could trigger situations of drought in the studied region. These observations can be juxtaposed with radiative flux during dry periods, resulting in elevated atmospheric aerosol values, particularly between LN-0708 and LN-1011. The study by [39] showcases that radiative forcing efficiency due to scattering (EF-AOD) during dry periods demonstrates significant impacts on radiative flux, notably in forested areas, due to increased scattering caused by biomass burning. These findings are in accordance with the research by [52], identifying that forest fires generate particles and gases, diminishing radiative flux by up to 50%.

Increases in AOD and EAOD-BC can be attributed, as evidenced by Liu et al. [53], highlighting that aerosol radiation interaction (ARIs) from biomass burning tends to suppress low-level liquid clouds, promoting local warming, increased evaporation, and favoring solid clouds at higher altitudes, resulting in condensation at higher levels. These ARIs findings are in line with the ωA demonstrated in this article, where low-level ωA values (indicating decreased upward motion) were observed across virtually the entire studied region during La Niña periods (LN-0708 and LN-1011). In contrast, during El Niño periods, ωA values exhibited different behavior, fluctuating between positive and negative.

Marengo et al. [54] argue that, during El Niño periods, droughts cannot always be solely attributed to it. As noted by the authors, in 2005, other influences such as conditions in the Tropical Atlantic Ocean, reduced moisture transport from the northeast trade winds to the southern Amazon, and decreased atmospheric convection also played significant roles. It is worth noting that, even during El Niño periods, above-average precipitation is possible in the Amazon region. These factors partly account for the low aerosol load values recorded in the EN-0607 dataset. In the preceding years, 2004 and 2005, characterized by prolonged dry spells as observed by Marengo et al. [54] in the Amazon, there was a substantial increase in forest fire incidents. However, in the years 2006 and 2007, especially during the

timeframe covered by EN-0607, the oceanic conditions in the Atlantic, such as the Dipole phase, created circumstances that did not favor the typical El Niño-induced droughts.

## 5. Conclusions

This study delved into the relationship between ENSO (El Niño Southern Oscillation) and atmospheric aerosols in the Legal Amazon region, employing metrics such as AOD (Aerosol Optical Depth) and EAOD-BC (AOD Extinction for Black Carbon) at AERONET sites in Alta Floresta, Ji-Paraná, and Rio Branco between 2006 and 2011. It also assessed the anomaly of specific humidity (QA) for different ENSO classes. During El Niño events, nearly 100% positive QA values were observed. In contrast, pronounced negative values were identified in the early months of La Niña influence.

Statistical analyses unveiled a significant relationship between ENSO and aerosols. The findings indicated lagged correlations of 4 to 6 months between ENSO and AOD values, physically influenced by the Walker circulation, meteorological elements, and oceanic conditions, aligning with earlier discoveries by Kumar and Hoerling [44], Alexander [45], and Marengo [54]. Following El Niño episodes, during subsequent La Niña events, there was a sharp increase of over 100% in AOD, especially in Alta Floresta and Ji-Paraná—an occurrence considered unusual for the Legal Amazon region.

In essence, this study discloses an association between ENSO and aerosol loads in the Legal Amazon, demonstrating the effects of these climatic phenomena on regional atmospheric dynamics, bearing significant implications for local weather and environmental conditions, particularly during transitions between El Niño and La Niña. Future ENSO events hold the potential to modulate aerosol loads in the Legal Amazon. Given the identified association, it is crucial to monitor how upcoming El Niños and La Niñas will impact AOD and EAOD-BC in the region, especially considering temporal delays and changes in the Walker circulation, providing essential approaches to predict and mitigate environmental impacts associated with atmospheric dynamics and air pollution.

**Supplementary Materials:** The following supporting information can be downloaded at: https://www.mdpi.com/article/10.3390/cli12020013/s1, Figure S1: Time series of aerosol optical depth (AOD) at 550 nm from AERONET with monthly average and monthly standard deviation for the sites of (a) Alta Floresta, (b) Ji Paraná, and (c) Rio Banco sites; Figure S2: Time series with the monthly accumulated fire outbreaks for the states where the AERONET sites are located. (a) Acre, where the Rio Branco site is located; (b) Mato Grosso, where the Alta Floresta site is located; and (c) Rondônia, where the Ji Paraná site is located. The y-axis of the graph is displayed on a logarithmic scale. Fire outbreak data derived from the National Institute for Space Research (INPE), available at: http://terrabrasilis.dpi.inpe.br/queimadas/bdqueimadas/#exportar-dados; Figure S3: Cross-section between longitude and pressure levels (hPa) within the longitude range of 10° and −10° for the average vertical velocity (omega) in Pa/s between the period from 09/2006 to 01/2007, covering the EN-0607 period; Figure S4: Cross-section between longitude and pressure levels (hPa) within the longitude range of 10° and −10° for the average vertical velocity (omega) in Pa/s between the period from 07/2007 to 06/2008, covering the LN-0708 period; Figure S5: Cross-section between longitude and pressure levels (hPa) within the longitude range of 10° and −10° for the average vertical velocity (omega) in Pa/s between the period from 08/2009 to 05/2010, covering the EN-0910 period; Figure S6: Cross-section between longitude and pressure levels (hPa) within the longitude range of 10° and −10° for the average vertical velocity (omega) in Pa/s between the period from 07/2010 to 06/2011, covering the LN-1011 period.

**Author Contributions:** Conceptualization, A.G.C.P., R.P., P.C.R.S. and R.V.S.P.; methodology, A.G.C.P., R.P., G.C. and B.I.; data curation, data processing and analysis, A.G.C.P., R.P., P.C.R.S. and R.V.S.P.; interpretation of the analysis results, A.G.C.P., R.P., P.C.R.S., R.V.S.P., G.C. and B.I.; formal analysis, A.G.C.P., R.P., R.V.S.P. and G.C.; writing—original draft preparation, A.G.C.P.; writing—review and editing, A.G.C.P., R.P., P.C.R.S., R.V.S.P., G.C. and B.I.; visualization, A.G.C.P., R.P., P.C.R.S., R.V.S.P., G.C. and B.I.; supervision, R.P., G.C. and B.I.; project administration, A.G.C.P., R.P. and G.C.; funding acquisition, R.P. All authors have read and agreed to the published version of the manuscript.

**Funding:** This work was supported by UFPA ("Universidade Federal do Pará"; Federal University of Pará) under the projects PRO4463-2020, PRO3906-2019, and the PAPQ ("Programa de Apoio à Publicação Qualificada"; Qualified Publication Support Program).

**Data Availability Statement:** Data Availability Statement: All the data used in this study are publicly available for free. Therefore, please refer to Section 2 for the sources in the materials.

**Acknowledgments:** The authors express their gratitude for the financial support provided by the PAPQ ("Programa de Apoio à Publicação Qualificada"; Qualified Publication Support Program) affiliated with UFPA ("Universidade Federal do Pará"; Federal University of Pará).

**Conflicts of Interest:** The authors declare no conflicts of interest.

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
