# Peer review of "Relationship between El Niño-Southern Oscillation and Atmospheric Aerosols in the Legal Amazon"

_climate, doi:10.3390/cli12020013_

Round 1
Reviewer 1 Report
Comments and Suggestions for Authors
This manuscript has scientific relevance and uses a method well reported by the scientific community that addresses the topic. Therefore, I suggest acceptance.
Author Response
Dear Reviewer,
We, the authors of the study entitled "Relationship between El Niño-Southern Oscillation and Atmospheric Aerosols in the Legal Amazon," are profoundly pleased with your contributions in terms of review. Undoubtedly, your analysis has substantially elevated the level of this study. We have endeavored to adapt the manuscript more clearly and objectively, including some factors that significantly aid in understanding the study.
Reviewer 2 Report
Comments and Suggestions for Authors
Comments on the Quality of English LanguageMinor editing of English language is required in the manuscript.
Author Response
Dear Reviewer,
We, the authors of the study titled “Relationship Between El Niño-Southern Oscillation and Atmospheric Aerosols in Legal Amazon” are deeply pleased with your contributions in terms of review and suggestions. Undoubtedly, your review and suggestions have substantially raised the level of this study. We have endeavored to respond to your inquiries in the manuscript more clearly and objectively, including some factors that strongly aid in understanding the study. All text revisions, new figures, and analyses are enclosed in a ".zip" file under the "Reply to Academic Editor" tab.

Reviewer 3 Report
Comments and Suggestions for Authors
Good work. Based on up-to-date techniques and described in detail. Results are also fully presented and discussed in detail.
Only a minor typos have to be corrected: at line 206 "As described by Boker et al. [34]34," has to be corrected.
Author Response
Dear Reviewer,
We, the authors of the study titled "Relationship between El Niño-Southern Oscillation and Atmospheric Aerosols in the Legal Amazon," are profoundly pleased with your contributions in terms of review. Undoubtedly, your analysis has significantly enhanced the quality of this study. We have endeavored to tailor the manuscript in a clearer and more objective manner, incorporating certain factors that greatly assist in understanding the study. All text revisions, new figures, and analyses are enclosed in a ".zip" file under the "Reply to Academic Editor" tab.
Round 2
Reviewer 2 Report
Comments and Suggestions for Authors
The authors have addressed my comments. I have only two minor points.
1) What is the uncertainty range of the MERRA2 derived vertical velocity. How it will influence the findings?
2) If possible improve the quality of Figure 4 and 5
I recommend this article for publication after minor revision.
Author Response
Dear Reviewer,
I want to express my gratitude for your continued evaluations, which have proven invaluable to this study.
Beginning at line 214 (section 2.3), we've incorporated a brief analysis concerning the assessment of uncertainties in the vertical velocity of MERRA-2 associated with radar observational data. It's worth noting that there's a limited body of research on evaluating uncertainties in vertical velocity. However, the study by Uma et al. [34], which I referenced, adequately addresses these uncertainties.
Regarding Figures 5 and 6, I agree that they require adjustments. The axes need to be more apparent, and I've made them slightly thicker. Additionally, I've included a shaded background with blue and red colors to represent El Niño and La Niña, respectively.
As for the methodology section's improvements, there was a brief text initially present due to the Climate template, which we inadvertently overlooked. We've rectified this by making the necessary adjustments.
Color observations in the text:
Red: Corrections from Round 1 of review;
Blue: Corrections from Round 2 of review.